# Comparison of DEM Models with Different Elemental Dimensions for TBM Disc Cutter Rock Fragmentation

Chen Xu [1,2], Yujie Zhu [2,3], Danqing Song [2,*], Xiaogang Guo [1], Xiaoli Liu [2], Enzhi Wang [2] and Runhu Lu [4]

1 Changjiang Institute of Survey, Planning, Design and Research, Wuhan 430010, China
2 State Key Laboratory of Hydroscience and Engineering, Tsinghua University, Beijing 100084, China
3 Key Laboratory of Geotechnical Mechanics and Engineering of Ministry of Water Resources,
Wuhan 430010, China
4 School of Civil Engineering and Architecture, Henan University, Kaifeng 475004, China
* Correspondence: songdq2019@mail.tsinghua.edu.cn

**Abstract:** Disc cutters are the dominant tool used in the excavation of hard rock formations in any underground construction application, such as when tunneling using tunnel-boring machines (TBM), as well as in shaft- and raise boring operations. Optimization of the cutting geometry of a given disc cutter for application in a rock formation often involves full-scale cutting tests, which is a difficult and costly proposition. An alternative to full-scale testing is the numerical simulation of TBM disc cutters for optimization under different settings. Recent efforts in the field of numerical simulations of rock cutting have shown the relative success of discrete element models, such as particle flow code (PFC), to simulate two- and three-dimensional rock fragmentation. This study is focused on a sensitivity analysis of PFC simulation of rock-cutting relative to the size of the elements. The calculated cutting forces were compared with the recorded forces under various conditions during full-scale tests using a linear cutting machine (LCM) on Colorado red granite (CRG). The estimated cutting coefficient and specific energy in the LCM tests and simulations showed good correlations, which validates the numerical simulation results. Two- and three-dimensional models showed that two-dimensional numerical models can offer a qualitative assessment of crack development, whereas three-dimensional models could be used to estimate the specific energy when cutting. The results can help in predicting the cutting forces in different rocks and ultimately improving disc-cutter geometry and cutter-head design.

**Keywords:** disc cutters; tunnel-boring machine (TBM); discrete element modeling (DEM); cutting force; specific energy; linear cutting machine (LCM)

## 1. Introduction

Mechanized excavators, such as tunnel boring machines (TBMs), offer higher speed and greater safety in operations, and this has led to their widespread use in underground construction and tunneling projects. The basic mechanism for rock fragmentation in hard rock applications is the use of disc cutters. The cutting performance of the disc cutters directly influences the efficiency of rock cutting and the success of the operation [1], which, in turn, depends on the rock properties and cutter geometry (i.e., cutter diameter, tip width, disc cutter spacing, and penetration). Previous researchers [2–4] have discussed optimal spacing and its relationship to cutting efficiency, as represented by specific energy. While determining the optimal penetration and spacing of disc cutters is crucial for increasing the efficiency of the disc-cutting process, the traditional approach toward optimization of the cutting geometry has been through full-scale cutting tests, via the comparison of DEM models with different elemental dimensions for TBM disc cutter rock fragmentation.

There are two main approaches to determining the optimal penetration and spacing of disc cutters: (i) experimental tests, which include the full-scale linear cutting machine (LCM) test, full-scale rotary cutting machine (RCM) test, and small-scale TBM test; and

(ii) numerical simulation methods, such as the finite element method (FEM), finite difference method (FDM), discrete element method (DEM), boundary element method (BEM), and hybrid methods. In the experimental approaches, the LCM test is the most commonly used device for researching disc-cutter performance. Rostami [3,5] used the LCM test to develop the Colorado School of Mines (CSM) TBM performance prediction model. The LCM test can address the scale effect, and the results are directly applicable to assessing the TBM performance in an actual project. There are many studies using full-scale cutting tests that explore the optimization of cutting geometry for rock cutting by disc cutters. Gertsch et al. [6] conducted a series of LCM tests on Colorado red granite (CRG) and determined a specific disc-cutter spacing that provides close to optimum specific energy. Geng et al. [7] used the RCM test to research the cutting forces and specific energy of disc cutters in different rock types. The main advantage of experimental studies is that they provide a fundamental understanding of the cutting process and rock failure mechanisms under the given cutter and geometrical configurations. Although full-scale experimental tests, such as the LCM and RCM tests, offer many obvious advantages, they are expensive and time-consuming [4]. There are limited facilities and research institutions that have LCM or RCM testing equipment. Therefore, there has been some interest in the use of numerical simulation methods to model the rock-cutting process and conduct a sensitivity analysis instead of full-scale cutting tests. Previous researchers have usually designed new disc cutters based on laboratory tests and numerical simulation methods. For example, Kim et al. [8] proposed a new estimation method to optimize the TBM cutter-head drive design, based on full-scale tunneling tests. Sun et al. [9] designed a new layout for the disc cutters according to numerical models and the cooperative coevolutionary algorithm. Xia et al. [10] used a multi-objective and multi-geologic conditions optimization program to optimize the design of disc cutters.

Given the increased capabilities of computers, many researchers have used a variety of numerical methods to analyze rock fragmentation using disc cutters. Cho et al. [4] used a linear Drucker–Prager constitutive model with AUTODYN-3D finite element analysis software to simulate an LCM test. They summarized the actual chipping mechanism and developed a corresponding experimental platform. The use of numerical simulation systems that are based on modeling the rock as a continuum (FEM, BEM, FDM, etc.) does not offer good results, due to the fact that the process of rock-cutting creates cracks and fractures, compromising the basic assumption of these models that the medium is continuous. In general, the numerical models that are continuum-based cannot accurately handle large deformations and fragmentation processes [11]. Obviously, rocks are a collection of blocks and particles with many interfaces and even joints and fractures at the mesoscopic level. These joints and fractures of rock are difficult to simulate with an FEM or BEM, but they can easily be simulated with a discrete element model (DEM).

DEM requires substantial computational resources for simulations, which limits its applicability to modeling large-scale problems. However, it provides a framework to describe de-bonding (i.e., fracturing) among discrete elements that closely simulates the natural process of fracture propagation [12]. Cundall [13] used the discrete element theory to propose a method for calculating particle flow. The particle flow code (PFC) method is widely used in numerical simulation studies of rock mechanics tests, the excavation of underground spaces, rock-slope engineering, mining engineering, and various other geotechnical fields [14–16]. Regarding the simulation of rock fragmentation by disc cutters using DEM methods, many researchers selected different element dimensions to simulate the rock and disc cutters. Gong et al. [17,18] simulated the cutting process of rock mass by a TBM cutter, using a two-dimensional DEM model, and studied the effect of joint spacing and orientation on rock fragmentation. Li et al. [19] used PFC$^{2D}$ to simulate the process of TBM indentation. Choi et al. [20] analyzed the cutting power of a disc cutter in a jointed rock mass. Moon et al. [21] used DEM to study the optimal rock-cutting conditions of a hard rock TBM, based on a two-dimensional model. Gong et al. [22] used chip thickness and chipping area to determine the efficiency of TBM excavation. These DEM applications

mainly used a two-dimensional model to reduce the calculation time. Although a three-dimensional model is more realistic, it is not often used with a DEM because of the large amount of calculation required. A few researchers have employed three-dimensional DEM models to offer a more realistic simulation of the cutting process. Choi et al. [23] used the contact bond model to simulate the rock in PFC$^{3D}$ and analyzed the rock-cutting behavior, but this model is not suitable for hard rock because of its contact model. Bahr et al. [24] discussed some of the challenges facing the three-dimensional modeling of rock cutting. Wu et al. [25] established PFC$^{3D}$ models to analyze the relationship between the mean and peak force during cutting with a disc cutter. To analyze the differences between the two- and three-dimensional models, the current study focuses on comparing the impact of elemental dimensions, to provide a more realistic simulation of rock-cutting, as represented by comparing the results with measured normal forces and specific energy. Validation of the modeling and related parameters, including the optimal size of particles in PFC, allows for the development of the method of determining the optimal conditions for a TBM. The remainder of this paper is organized as follows. Section 2 presents the simulation models and methods, including the contact model and the optimal condition of the simulation models. The three-dimensional simulation results and specific energy of different disc-cutter spacings and penetrations are presented in Section 3. Section 4 compares the cutting force of the two- and three-dimensional models in PFC. Section 5 concludes this article.

## 2. Numerical Modeling

### 2.1. Disc Cutter Geometry

Disc cutters are usually made of high-strength steel, which is known for its high durability. After the late 1970s, constant cross-section (CCS) profile cutter rings replaced their V-shaped predecessors because of their durability and ability to maintain high cutting efficiency over an extended time [2]. For easy verification of the simulation results, we employed a wall model to simulate the disc cutters in PFC. This is reasonable because this study was focused on the rock fragmentation mechanism rather than on the durability of the cutter tip. A 432-millimeter (17-inch)-diameter disc cutter is shown in Figure 1. The three-dimensional (3D) model of the disc cutters was created with a 3D CAD system and then imported to PFC$^{3D}$ software as two walls. For the two-dimensional model, the disc cutter was simplified, and the tip width of the disc cutter was set to 13 mm to match the available data for the LCM tests. Figure 2 shows the two-dimensional disc cutter model.

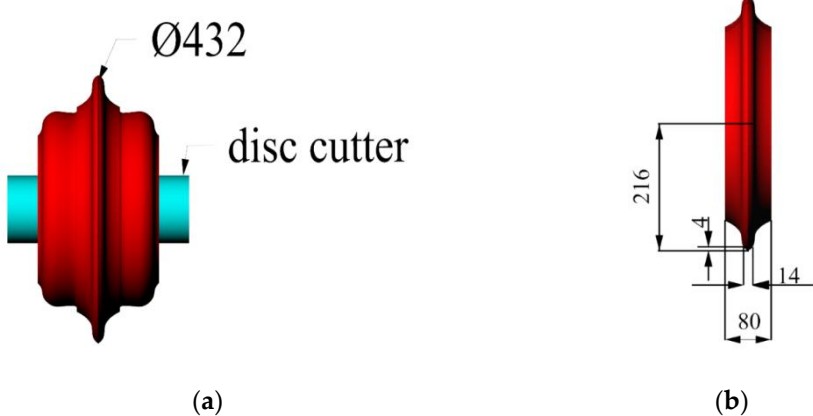

(**a**)          (**b**)

**Figure 1.** Three-dimensional disc cutter model (unit: mm): (**a**) geometric model of a disc cutter; (**b**) the CCS profile of the cutter used in this research (unit: mm).

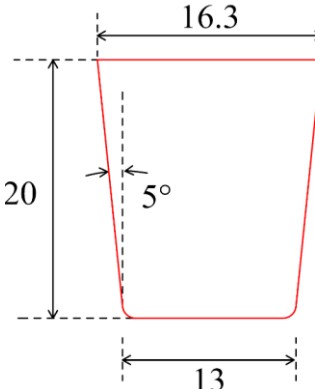

**Figure 2.** Two-dimensional disc cutter model (unit: mm).

*2.2. Modeling Rock Specimens*

During the rock-cutting process, the rock material experiences high stress concentration, yielding, fracture initiation and propagation, damage, and failure. The pertinent material properties of rock directly affect the cutting forces and load on the disc cutter. In this study, PFC software was used, as reported in [26], to simulate the rock specimen. Setting up the PFC model and the contact model are the most important parts of setting up the simulation, similar to when assigning the constitutive model in an FEM. In most cases, previous researchers [14,21,27,28] have selected a linear parallel bond model (LPBM) to simulate the rock's strength properties. However, a long-standing limitation of an LPBM is that if the unconfined compressive strength of a typical compact rock is matched, then the direct tensile strength of the model is too high [14]. This is why the LPBM is not suitable for a typically hard rock, such as granite. To overcome this limitation, a flat-jointed bonded-particle model was proposed that is based on the premise that a closer match to the structural and microstructural features will provide a closer match to its real macroscopic behavior [29]. In this research, the rock specimen selected for the analysis was CRG; therefore, a flat-jointed contact model (FJCM) was selected for the PFC simulations.

2.2.1. Flat-Jointed Contact Model

The FJCM can simulate the behavior of an interface between two notional surfaces, each of which is connected rigidly to a piece of the body [26]. The FJCM provides the macro behavior of a finite-sized, elastic, and bonded or frictional interface [29]. The interface is composed of elements that are individually bonded or unbonded. Figure 3 shows a schematic drawing of the mechanical behavior of FJCM [26].

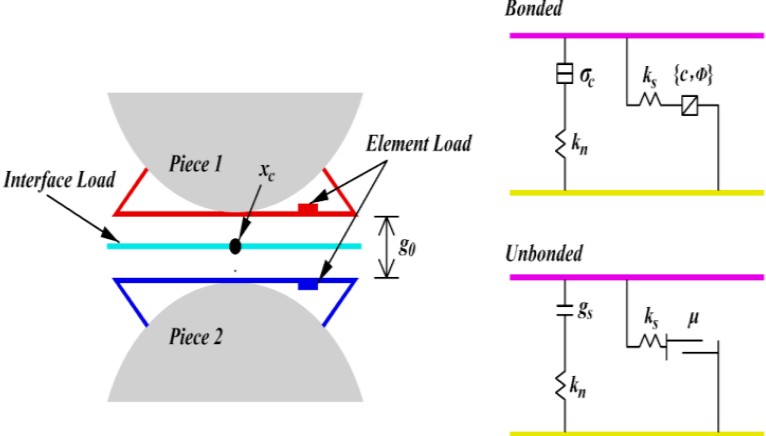

**Figure 3.** Schematic drawing of the mechanical behavior of FJCM.

The force–displacement law of the FJCM includes the contact force $\vec{F}$ and moment $\vec{M}$, which can be updated with the following equations. Each element carries a force $\vec{F^{(e)}}$ and moment $\vec{M^{(e)}}$:

$$\vec{F} = \sum_{\forall e} \vec{F^{(e)}} \tag{1}$$

$$\vec{M} = \sum_{\forall e} \left\{ \left( r^{(e)} \times \vec{F^{(e)}} \right) + \vec{M^{(e)}} \right\} \tag{2}$$

where $r^{(e)}$ is the relative position of element $e$.

The element force can be resolved into normal and shear forces, and the element moment can be resolved into twisting and bending moments [26]:

$$\vec{F^{(e)}} = -F_n^{(e)} \vec{n_c} + F_{ss}^{(e)} \vec{s_c} + F_{st}^{(e)} \vec{t_c} \tag{3}$$

$$\vec{M^{(e)}} = M_t^{(e)} \vec{n_c} + \vec{M_b^{(e)}} \tag{4}$$

where $F_n^{(e)}$ is the scalar value of the normal force; for signage, $F_n^{(e)} > 0$, $\vec{F_n^{(e)}}$ is tension and $F_n^{(e)} < 0$, $\vec{F_n^{(e)}}$ is compression. $\vec{n_c}$ is the direction of the normal vector. $F_{ss}^{(e)}$ is the component value of the shear force in the $\vec{s_c}$ direction and $F_{st}^{(e)}$ is the component value of the shear force in the $\vec{t_c}$ direction.

$M_t^{(e)}$ is the bending moment and $\vec{M_b^{(e)}}$ is the twisting moment.

The element normal and shear stresses can be calculated as:

$$\sigma^{(e)} = F_n^{(e)} / A^{(e)} \tag{5}$$

$$\tau^{(e)} = \| \vec{F_s^{(e)}} \| / A^{(e)} \tag{6}$$

where $\sigma^{(e)}$ is the normal stress, $\tau^{(e)}$ is the shear stress, and $A^{(e)}$ is the area of each element.

2.2.2. Mesoscopic Parameter Calibration

There is no straightforward method currently available for calibrating synthetic rock material according to its microscopic properties. The mesoscopic mechanical parameters need to be determined for the numerical simulation of particle flow and are usually obtained empirically [28,30].

The mechanical parameters of rock specimens include the uniaxial compressive strength (UCS), Brazilian tensile strength (BTS), elastic modulus (EM), Poisson's ratio, and shear strength. Three sets of UCS tests and BTS tests were conducted on CRG in the Earth Mechanics Institute's rock mechanics testing laboratory. The average values of the UCS, BTS, and EM in the PFC$^{3D}$ were determined and are summarized in Table 1. Therefore, the macroscopic parameters of CRG selected for this study were the UCS, BTS, and EM.

The FJCM is defined by the parameters of both the particle and contact models. For convenience in the numerical analysis, the following assumptions were taken from the literature [31,32]:

(1)   For the two and three-dimensional models, the minimum and maximum particle radii were 1.5 and 2 mm, respectively. The particles were evenly distributed by radius.
(2)   The radius multiplier of the FJCM is 1.0.
(3)   The particle density is equal to the rock density (i.e., 2650 kg/m$^3$).

**Table 1.** Summary of the results of the rock mechanics testing and numerical simulation by PDF, including the calibration error rate of the macro parameters.

| Marco Test Parameters | UCS), (MPa) | BTS, Mpa | Elastic Modulus (EM), (Gpa) |
|---|---|---|---|
| Average Value of 3 tests | 178.5 | 8.9 | 21.8 |
| Simulation Results in PFC$^{3D}$ | 179.1 | 8.8 | 21.4 |
| Error Rate in PFC$^{3D}$ | 0.34% | 1.12% | 1.83% |
| Simulation Results in PFC$^{2D}$ | 177.8 | 9.0 | 22.5 |
| Error Rate in PFC$^{2D}$ | 0.39% | 1.12% | 3.21% |

Therefore, the mesoscopic parameters of the FJCM included the bond gap, effective modulus, effective normal-to-shear stiffness ratio, number of elements in the radial direction, number of elements in the circumferential direction, tensile strength, cohesion, friction angle, and friction coefficient. The macro parameters and mesoscopic parameters considered in this study are listed in Table 2.

**Table 2.** Mesoscopic parameters of FJCM in the two- and three-dimensional numerical models.

| Marco Parameters | Uniaxial Compressive Strength (UCS) | | | Brazilian Tensile Strength (BTS) | | Elastic Modulus (EM) | | |
|---|---|---|---|---|---|---|---|---|
| Mesoscopic Parameters | Bond Gap (fj_gap0) | Deformability Effective Modulus (fj_emod) | Effective Normal-to-Shear Stiffness Ratio(fj_krat) | Number of Elements in Radial Direction (fj_nr) | Number of Elements in Circumferential Direction (fj_nal) | Tensile Strength (fj_ten) | Cohesion (fj_coh) | Friction Angle (fj_fa) |
| PFC$^{3D}$ | 0.0 | $11 \times 10^9$ | 1.5 | 2 | 4 | $4.4 \times 10^6$ | $38 \times 10^6$ | 30 |
| PFC$^{2D}$ | 0.0 | $27 \times 10^9$ | 1.5 | 4 | \ | $13 \times 10^6$ | $150 \times 10^6$ | 30 |

The construction of the UCS and BST numerical simulation models can be divided into the following steps.

Step 1: Use the "wall" command to enclose a Brazilian test disc, then generate randomly distributed spherical particles in this disc.

Step 2: Obtain the initial isotropic stress conditions of the particle assembly by expanding the radius of the particles.

Step 3: Delete the floating particles. A floating particle has a coordination number of less than 4.

Step 4: Assign the FJCM according to the mesoscopic parameter, based on the performed calibration runs.

Step 5. Finally, construct the numerical simulation models.

After many trials and comparisons, the stress–strain curves of the UCS and BTS of the three-dimensional model were derived, as shown in Figure 4. The final mesoscopic parameters of the FJCM were determined, as given in Table 1, with the best results matching the laboratory tests. The calibration error rate of the macro parameters of the rock is presented in Table 1.

The mesoscopic parameters of the three- and two-dimensional models are slightly different, so some rebuild simulation models were developed and tested. After a series of trials, the UCS, BTS, and EM for the two-dimensional model were also determined and are given in Table 1. Table 2 presents the final mesoscopic parameters of the FJCM. Figure 5 shows the stress–strain curves of the UCS and BTS of the two-dimensional model.

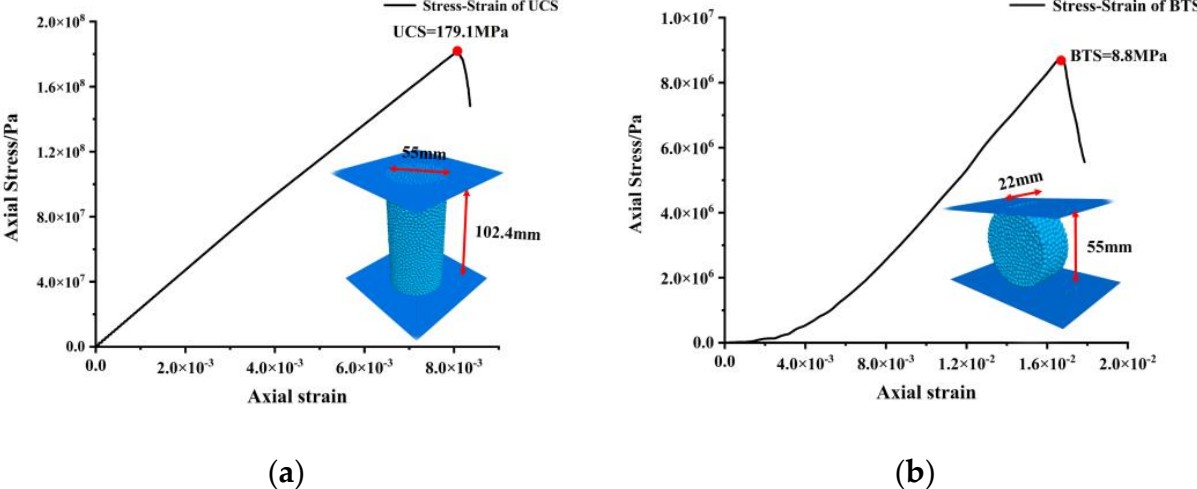

**Figure 4.** Stress–strain curves and three-dimensional simulation model of UCS and BTS: (**a**) UCS; (**b**) BTS.

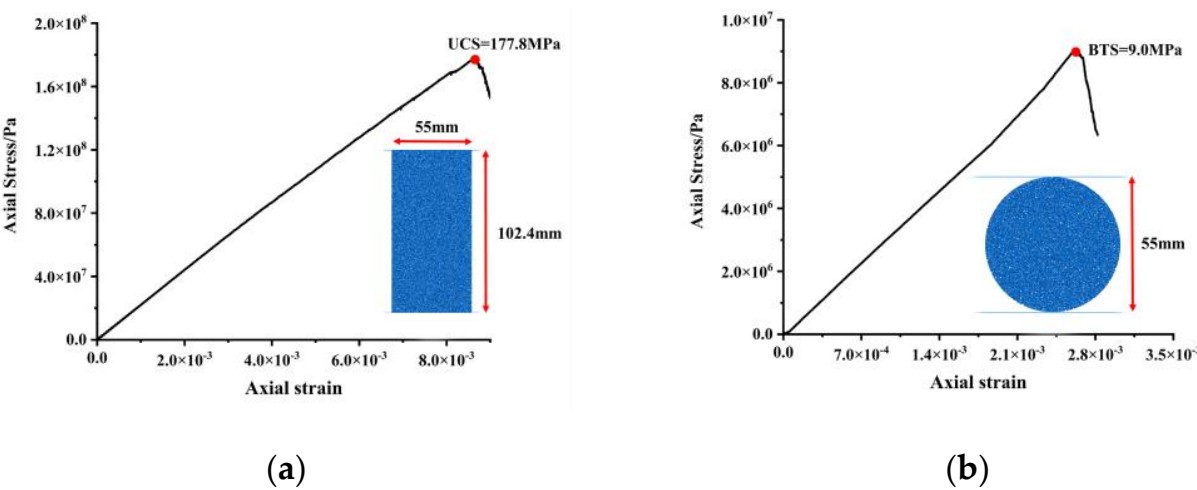

**Figure 5.** Stress–strain curves and two-dimensional simulation model of UCS and BTS: (**a**) UCS; (**b**) BTS.

### 2.3. Development of Numerical Models of Rock-Cutting

During the rock excavation process when using a disc cutter, the geological conditions in the direction of advance are complex and destined to change; various factors influence the efficiency of TBM tunneling. Therefore, it is impossible to include and accurately simulate all the geological factors impacting the rock-cutting process. Figure 6 shows the mechanism of rock fragmentation with disc cutters. Tensile cracks are caused by the induced stresses from the disc cutters, and these tensile cracks continue to reach the cutting surface or intersect another crack created by cutting during previous passes. If one or more tensile cracks meet, or if the cracks reach the free surface, chipping occurs [3]. However, the actual cutting is a three-dimensional process, as shown in Figure 7.

PFC$^{3D}$ was used to construct three-dimensional models of the disc cutters and the rock specimen. The rock specimen had dimensions of 300 mm × 200 mm × 100 mm and consisted of 241,135 particles. Because the disc-cutter wear was outside the scope of this study, we used a wall model to simulate the disc cutters. We used the principle of relative motion to simulate the process of the disc-cutting of rock. The geometries and boundary conditions used in the numerical model are shown in Figure 8. The rock specimen model was fixed in all directions, and the disc cutter model moved in the horizontal direction (*y*-direction) with a constant velocity for linear cutting. Simultaneously, the cutter was fixed

to maintain a given penetration depth in the vertical direction (*z*-direction) and was rotated at a constant angular velocity (*x*-direction). To match the LCM cutting conditions, two disc cutters that were cutting in sequence were modeled. For comparison with the LCM test results, six sets of numerical simulations with disc-cutter spacings of 62.5 and 75 mm and penetrations of 3.2, 4.4, and 6.4 mm were modeled.

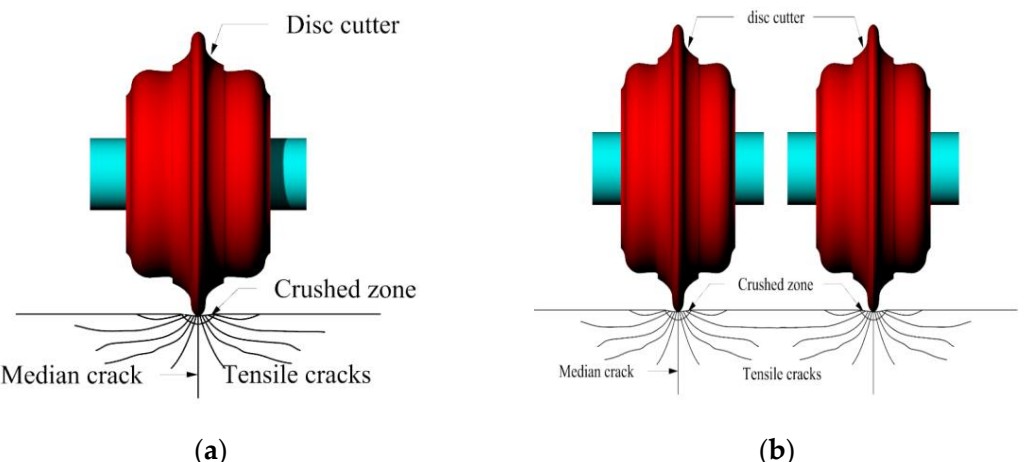

(**a**)                                                     (**b**)

**Figure 6.** Development of crushed zones and cracks when using TBM disc cutters. (**a**) single disc cutter, (**b**) double disc cutters.

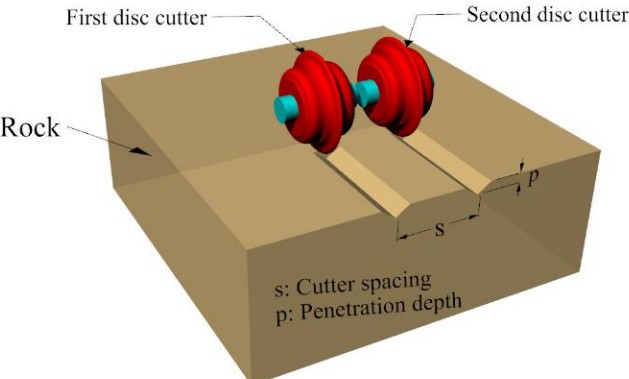

**Figure 7.** Three-dimensional rock-cutting geometry.

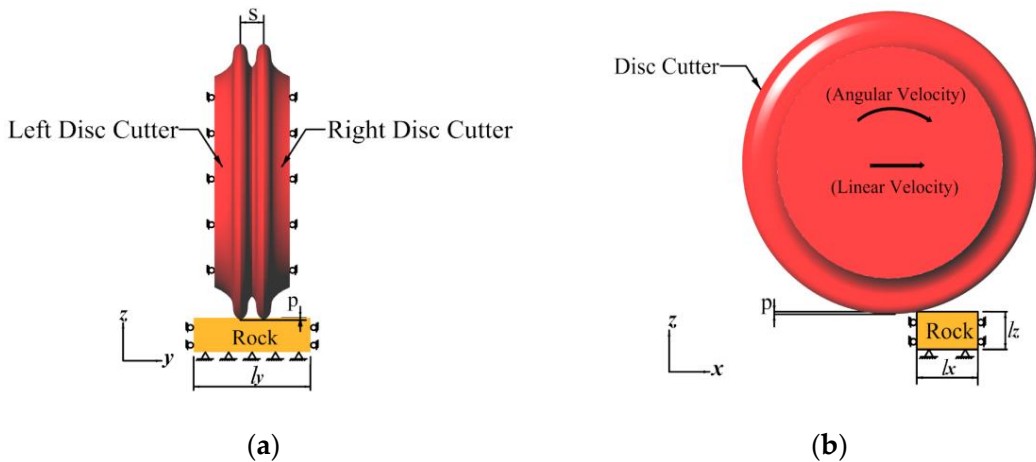

(**a**)                                                     (**b**)

**Figure 8.** Geometries and boundary conditions used in the numerical model (unit: mm): (**a**) side view; (**b**) front view.

*2.4. Determining the Optimal Conditions for Disc-Cutter Rock Fragmentation*

The specific energy (SE) is defined as the required energy to cut a single unit volume of rock. This is one of the primary performance indicators used in an assessment of the efficiency of an operation in mechanical excavations and likewise applies to the disc cutter. SE is used to determine the efficiency of machine performance as a function of geometrical parameters [2]. To determine the optimal conditions for rock cutting, SE needs to be minimized. SE can be calculated under various conditions, as follows:

$$SE = \frac{E_{total}}{V_{cut}} = \frac{MRF \times l_y + MNF \times p}{V_{cut}} \tag{7}$$

where $E_{total}$ is the total energy consumption during the cutting process, $V_{cut}$ is the cutting volume, $MRF$ is the mean rolling force, $l_y$ is the cutting distance, $MNF$ is the mean normal force, and $p$ is the penetration. In this study, we calculated the cutting volume from the number of eliminated particles after cutting. If the contact number of a particle was less than 4, then the particle was assumed to be cut off and was deleted.

## 3. Simulation Results of Three-Dimensional Models

### 3.1. Effect of Cutting Velocities on the Simulation Results

The moving speed of the disc on the rock in a numerical simulation model differs from reality, sometimes greatly, because in the numerical model, the speed is applied to the model through the time steps. In the actual LCM tests, the linear velocity was 254 mm/s (10 in/s), and the angular velocity of the disc was 1.18 rad/s. To determine the speed in the three-dimensional numerical model, five sets of simulations were conducted, and we set the linear velocity to 25.4, 12.7, 2.54, and 1.27 m/s. The selected spacing was 62.5 mm with a penetration of 3.2 mm. Figure 9 shows the calculation times and simulation results at different cutting speeds. All the simulations were performed on a desktop computer with the following technical parameters: one CPU with an Intel core (i7-7700K, with 4.20 GHz) and eight core processors, one RAM of 32 GB, and the 64-bit system type.

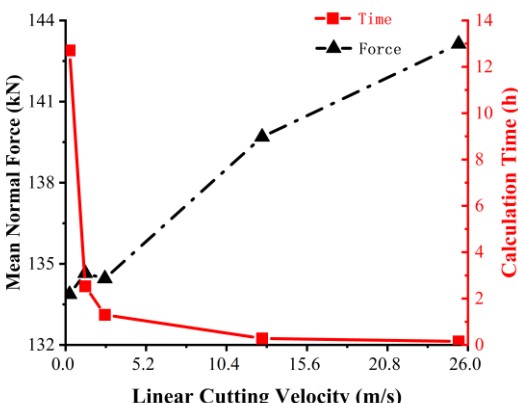

**Figure 9.** Calculation times and normal forces at various cutting speeds in the three-dimensional numerical simulation.

The results indicate that the calculation time decreased exponentially with increasing linear speed, while the normal force did not change substantially as the linear speed decreased. To optimize the tradeoff between the calculation time and accuracy, we selected a linear speed of 2.54 m/s for the numerical simulation model, where the corresponding angular velocity was 11.8 rad/s.

During the cutting process, the disc cutters were set up in the model as walls in PFC$^{3D}$, so we could record the wall forces in different directions. The two disc cutters are parallel to the stratification in the numerical simulation tests. Figures 10–13 show the normal and rolling forces for different spacings. Table 3 presents the mean normal force and rolling force under various conditions.

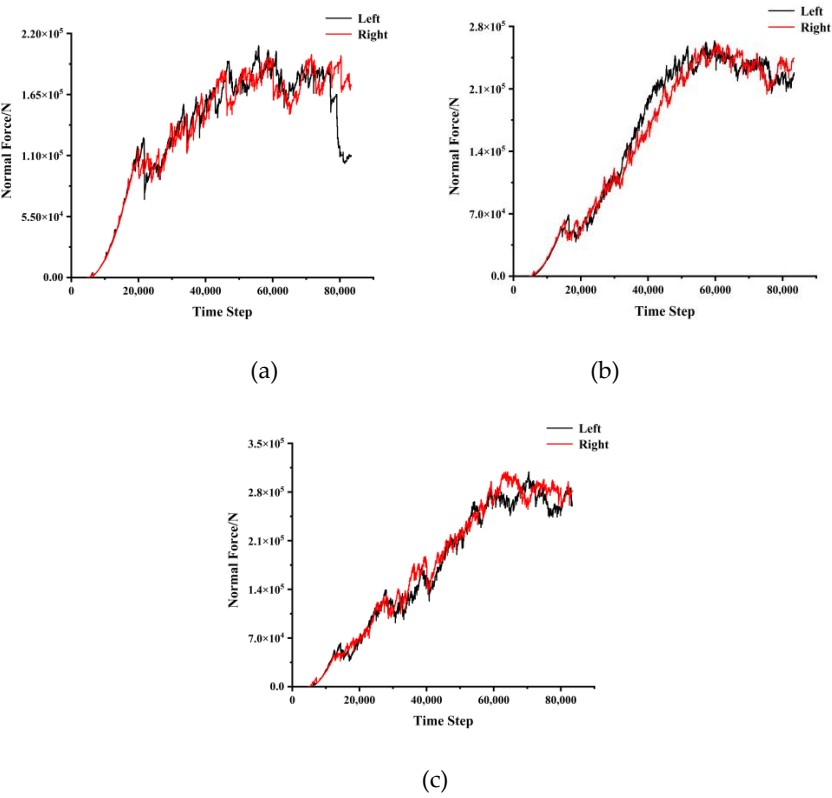

**Figure 10.** Normal force with a disc-cutter spacing of 62.5 mm: (**a**) penetration of 3.2 mm; (**b**) penetration of 4.4 mm; (**c**) penetration of 6.4 mm.

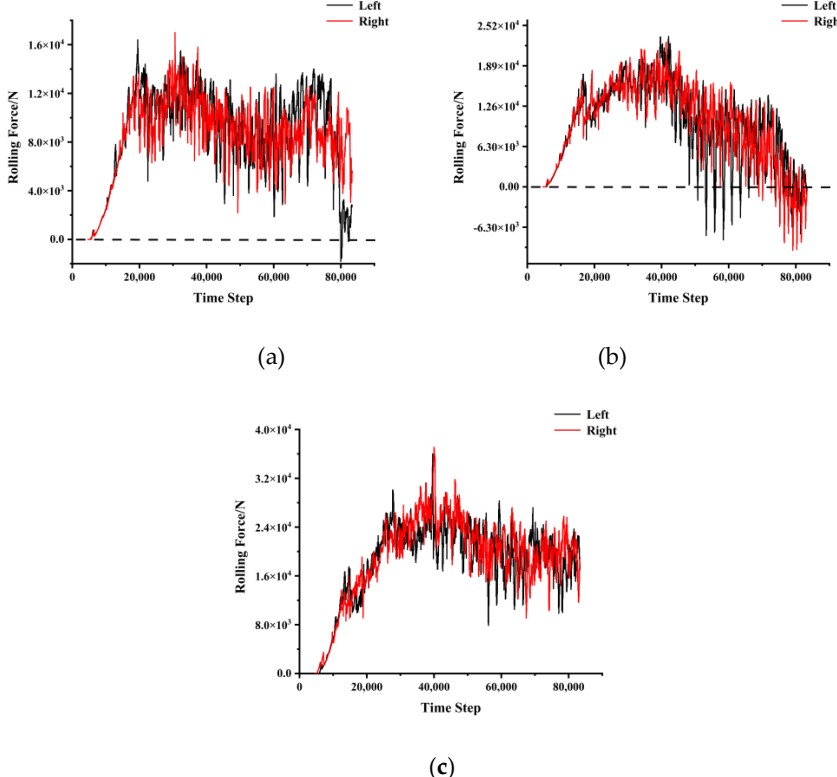

**Figure 11.** Rolling force with a disc-cutter spacing of 62.5 mm: (**a**) penetration of 3.2mm; (**b**) penetration of 4.4 mm; (**c**) penetration of 6.4 mm.

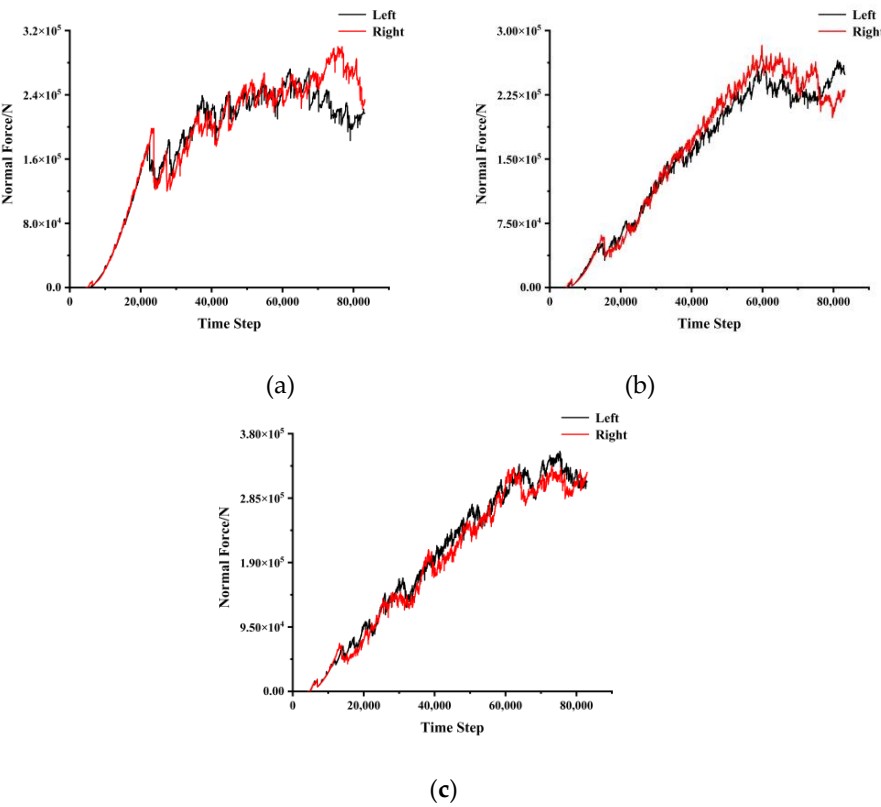

**Figure 12.** Normal force with a disc-cutter spacing of 75 mm: (**a**) penetration of 3.2 mm; (**b**) penetration of 4.4 mm; (**c**) penetration of 6.4 mm.

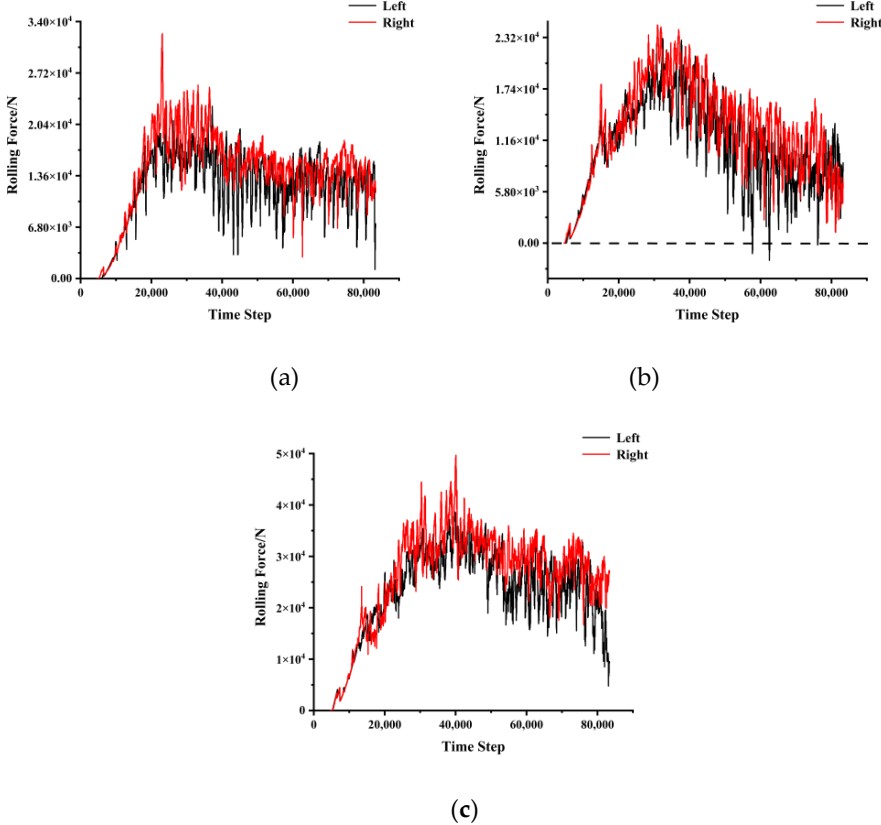

**Figure 13.** Rolling force with a disc-cutter spacing of 75 mm: (**a**) penetration of 3.2 mm; (**b**) penetration of 4.4 mm; (**c**) penetration of 6.4 mm.

**Table 3.** Results of the three-dimensional numerical models and LCM tests.

| Spacing (mm) | Penetration (mm) | PFC$^{3D}$ | | | LCM Tests | | | Error Rate | | |
|---|---|---|---|---|---|---|---|---|---|---|
| | | Mean Normal Force (kN) | Mean Rolling Force (kN) | Specific Energy (MJ/m$^3$) | Mean Normal Force (kN) | Mean Rolling Force (kN) | Specific Energy (MJ/m$^3$) | Mean Normal Force | Mean Rolling Force | Specific Energy |
| 62.5 | 3.2 | 129.14 | 6.10 | 22.93 | 128 | 7 | 37.80 | 0.89% | 12.86% | 39.34% |
| | 4.4 | 199.69 | 7.95 | 35.92 | 139 | 11 | 39.96 | 43.66% | 27.73% | 10.11% |
| | 6.4 | 196.08 | 13.17 | 38.43 | 155 | 18 | 45.72 | 26.50% | 26.83% | 15.94% |
| 75 | 3.2 | 130.04 | 7.48 | 27.53 | 186 | 12 | 52.56 | 30.09% | 37.67% | 47.62% |
| | 4.4 | 205.61 | 8.79 | 34.48 | 144 | 12 | 37.44 | 42.78% | 26.75% | 7.91% |
| | 6.4 | 196.33 | 15.84 | 42.06 | 186 | 25 | 53.28 | 5.55% | 36.64% | 21.06% |

A closer examination of the rolling forces shows that the negative values have been calculated close to the end of the run, which means that the direction of the rolling force changed. When the rolling force drops to 0, the disc cutters continue to move forward but there is no rock to cut. This process is very short in real tests, while in a numerical simulation, it needs to happen gradually through the time steps. The change in direction in terms of the force and recording of the negative value could be due to the inertia built into the disc-cutter motion. Meanwhile, since the cutter was still in contact with the rock, the normal force did not change to negative values. Therefore, when we calculate the mean values of the normal or rolling forces, this issue has to be taken into account.

*3.2. Comparison of the Simulated Forces with the LCM Results*

To verify the results of the numerical simulation, the results of the numerical analysis were compared with those of the full-scale rock-cutting tests on an LCM unit at the Earth Mechanical Institute (CSM). The LCM of CSM is shown in Figure 14. The recorded mean normal force, rolling force, and SE during the LCM tests are presented in Table 3.

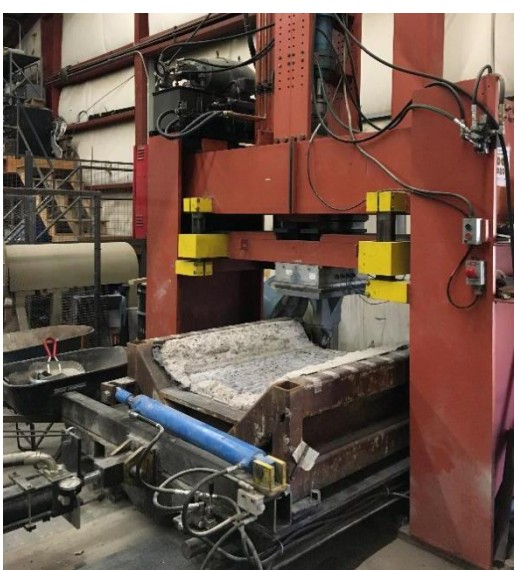

**Figure 14.** The linear cutting machine at the Earth Mechanical Institute (CSM).

Figures 15 and 16 show the mean normal force and mean rolling force in the simulations and LCM tests as a function of penetration. In terms of the mean normal force, the simulation results were generally higher than the results of the LCM tests. When the penetration was 4.4 mm, the normal forces of the simulation results were much higher than those in the LCM tests. There could be two reasons for this observation. First, in the numerical models, the two disc cutters were simulated together, while the actual testing on the LCM was on a single disc. In the LCM tests, after the first pass, the rock is damaged,

and its resistance to cutting is decreased. This decrease is known as conditioning and the disc is tracked to exploit these induced, new cracks for rock cutting in the following passes. This is not the case for the simulated cuts since the model assumed that there were no existing fractures or cracks in the rock. Second, there may have been measurement errors in the LCM tests for the given test settings. In LCM tests, the mean forces were obtained via three sets of experiments. For 75mm spacing and in the case of penetration, the measured mean normal forces were 191 kN and 100 kN.

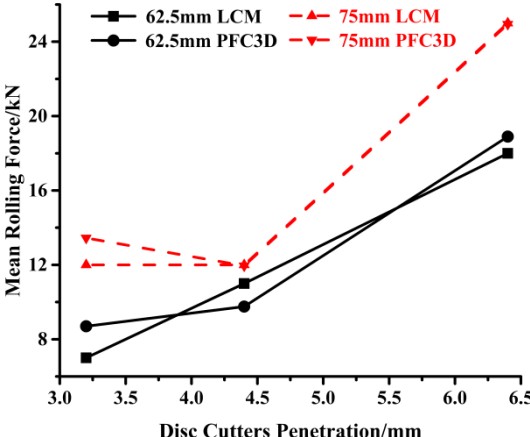

**Figure 15.** Comparison between the mean normal forces of the LCM tests and numerical simulations.

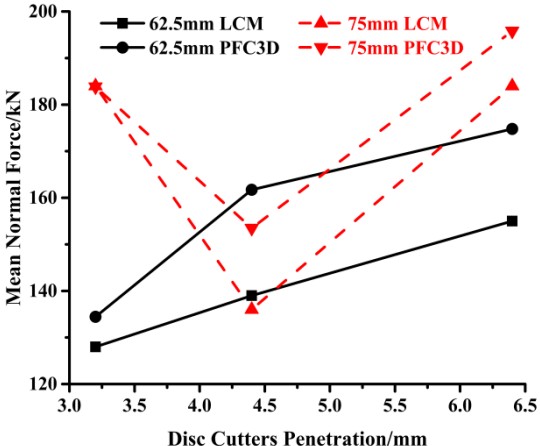

**Figure 16.** Comparison between the mean rolling forces of the LCM tests and numerical simulations.

The results of numerical simulations showed that the rolling forces were less than those measured by LCM tests. In the LCM tests, the rolling forces were recorded by a load cell, while in the numerical models, the disc cutter was modeled as a wall and the rolling force was recorded by means of the wall's reaction force. Therefore, different measurement methods could produce measurement errors.

### 3.3. Disc Cutters' Performance Comparison

The SE is defined as the required energy for a single unit volume of rock, which is often used to determine the optimal conditions for rock excavation. Figure 17 compares the SE results of the LCM tests and numerical simulations. At the same spacing-to-penetration ratio, the SE showed the same trend, while the SE was slightly lower when calculated by the PFC$^{3D}$ simulation than when measured in the LCM tests. This agrees with the results of Choi and Lee [23].

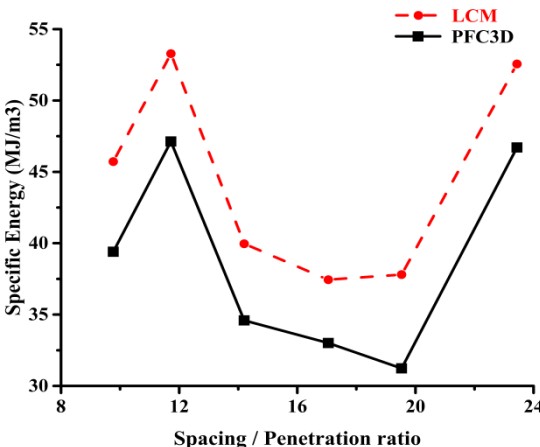

**Figure 17.** SE of various spacing-to-penetration ratios.

The reason for the lower SE calculated by $PFC^{3D}$ is the difference between the two calculations of excavated volume. On the one hand, in LCM testing, SE is calculated based on spacing and penetration, thereby assuming full clearance between the lines and nominal excavation volume, which is true as the cuts proceed and the rock is removed. On the other hand, the rock's calculated excavate volume in the simulation is based on the volume estimated to have been removed from the main sample. This warrants a closer look at the way that the excavated volume is estimated in the simulation; it could be that the values that are calculated are slightly larger than those of the nominal excavation volume used in the calculation of the SE in LCM tests.

The optimal value of the SE is a range that can be determined by observing the trend and demarcation of those S/P ranges that result in lower SE values.

## 4. Discussion

The two-dimensional numerical model is a reproduction of the disc-cutter penetration test and can also be used to determine rock fragmentation when using TBM disc cutters. With the development of detection methods, researchers can detect and analyze the whole process of hob penetration by means of acoustic emission, electronic speckle interference, digital image detection, and so on. Chen et al. [33] used acoustic emission and electronic speckle interference to monitor the process of disc cutter penetration into granite and sandstone, studying the influence of confining pressure on penetration force and crack propagation. Zhang et al. [34] tested the formation and expansion of cracks in the whole process of disc cutter penetration into rock, based on the digital image correlation method, and analyzed the formation mechanism of intermediate cracks. Liu et al. [35] analyzed the fracture behavior of TBM disc cutter penetration in different jointed rock masses using a digital image method. In this section, a comparison between the cutting forces estimated, based on numerical simulation, using two- and three-dimensional models in PFC was conducted to see the impact of modeling in two and three dimensions on the results. Figure 18 shows a screenshot of the two-dimensional numerical model in PFC. A rock specimen with dimensions of 400 mm × 200 mm was used for modeling where the discs were pressed to penetrate the rock. Overall, six sets of numerical simulations, with spacings of 62.5 and 75 mm and penetrations of 3.2, 4.4, and 6.4 mm, were conducted.

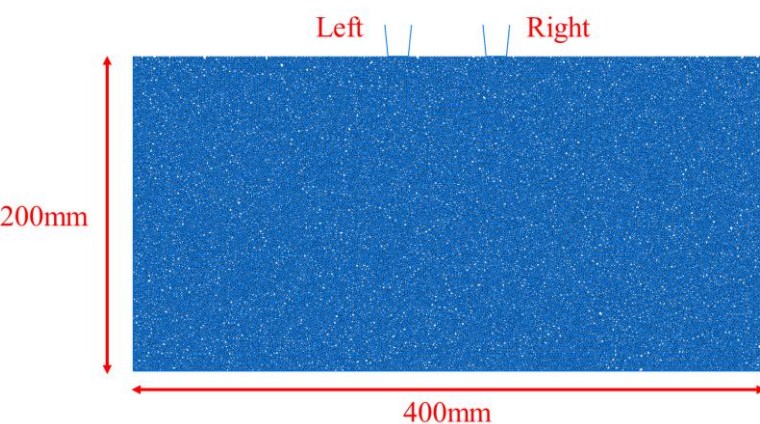

**Figure 18.** Screenshot of the two-dimensional numerical simulation model in PFC$^{2D}$.

To determine the speed in the two-dimensional numerical model, a series of five simulations with a penetration velocity of 1, 0.5, 0.1, 0.05, and 0.01 m/s were run. The examination of the effect of speed assumed a spacing of 62.5 mm and a penetration of 3.2 mm. Figure 19 shows the results for the calculation time and mean normal force at different speeds. To optimize the tradeoff between the calculation time and accuracy, we set the penetration speed for the follow-up simulations at 0.1 m/s.

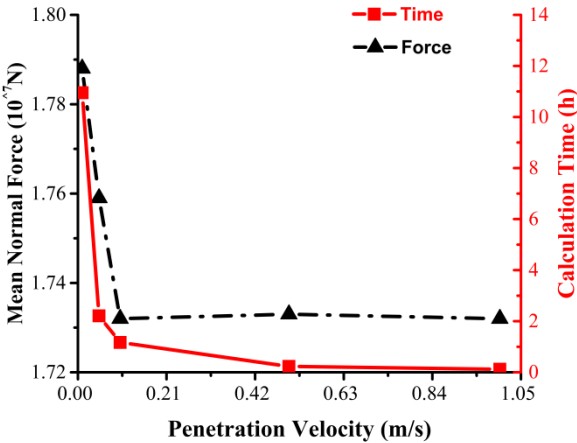

**Figure 19.** Calculation time and normal force for the two-dimensional numerical simulation at various penetration speeds.

The normal forces of the disc cutters were recorded during the indentation process and Figure 20 shows the results at a spacing of 62.5 mm. When the penetration was 4.4 or 6.4 mm, the normal force first reached the maximum value and then dropped. This shows that the rock specimen was damaged when the penetration was greater than 4.4 mm and that a crack and subsequent chips were formed, meaning that the pressure in the crushed zone or pressure bubble was released. However, the force should not, logically, then drop to zero as was observed in the simulations, since there is still contact, and some pressure is acting on the disc. The build-up of the pressure and the drop in pressure, which forms a saw-tooth pattern, is very common in the punch test.

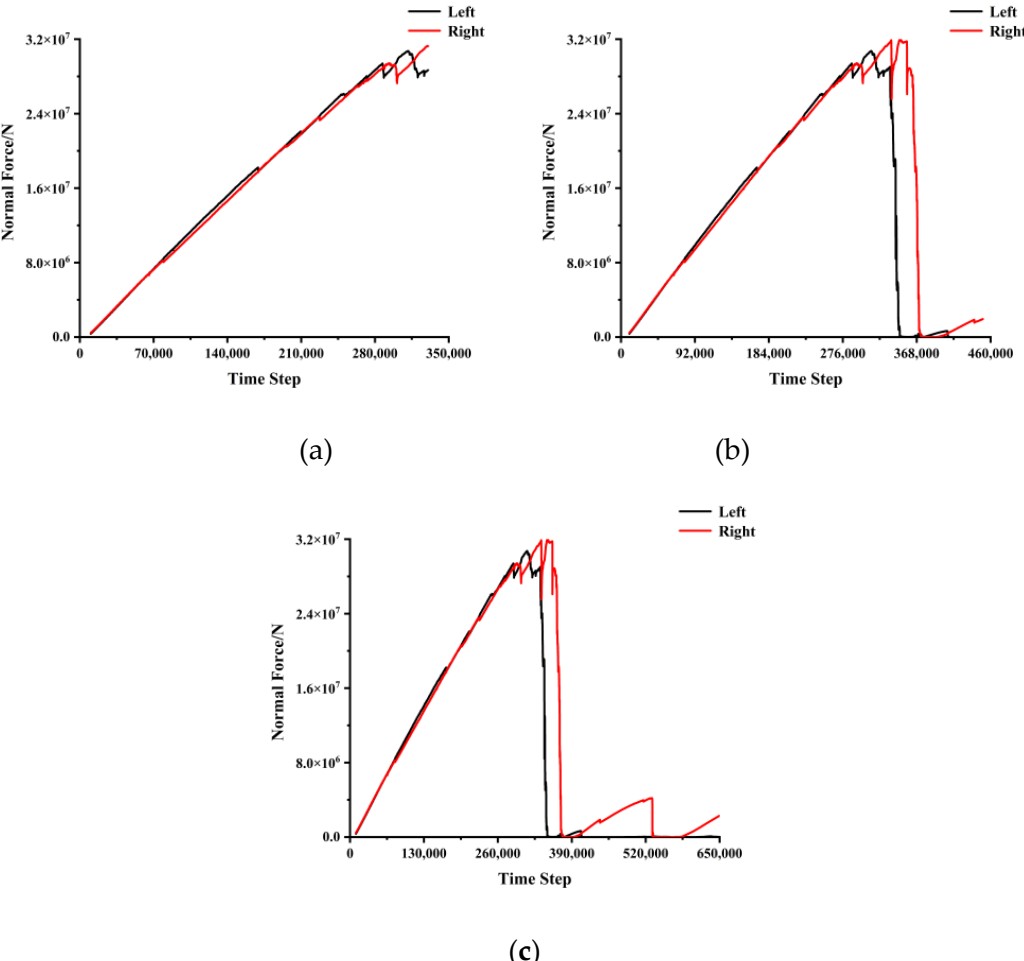

**Figure 20.** Variations of normal force as a function of penetration, at a spacing of 62.5 mm: (**a**) penetration of 3.2 mm; (**b**) penetration of 4.4 mm; (**c**) penetration of 6.4 mm.

The normal force calculated in the two-dimensional models was greater than the measured values in the LCM tests. This is because the mechanisms of the two methods were different. In the LCM tests, the rock specimens were cut with gradually increasing penetration. In the two-dimensional numerical simulations, the rock was pressed in a similar way to the punch penetration test, but by assuming the indentation by two relatively dull disc cutters. In addition, the hidden assumption in the two-dimensional model is the continuation of the disc tip profile in the third dimension, which, in reality, is not true. Therefore, the two-dimensional numerical model could not simulate the LCM tests and so determining the optimal cutting geometry using the two-dimensional model would be problematic. According to Figure 21, the fracture propagation pattern was different for the different penetrations. Thus, working from the simulation of the crack propagation pattern could only offer a qualitative assessment of the cutting process and not a quantitative measure of the cutting efficiency. However, the three-dimensional numerical model was more accurate and could offer an assessment of cutting conditions when comparing different disc-cutter geometries or different rock types, and could help in the calculation of SE to determine the optimal cutting geometry.

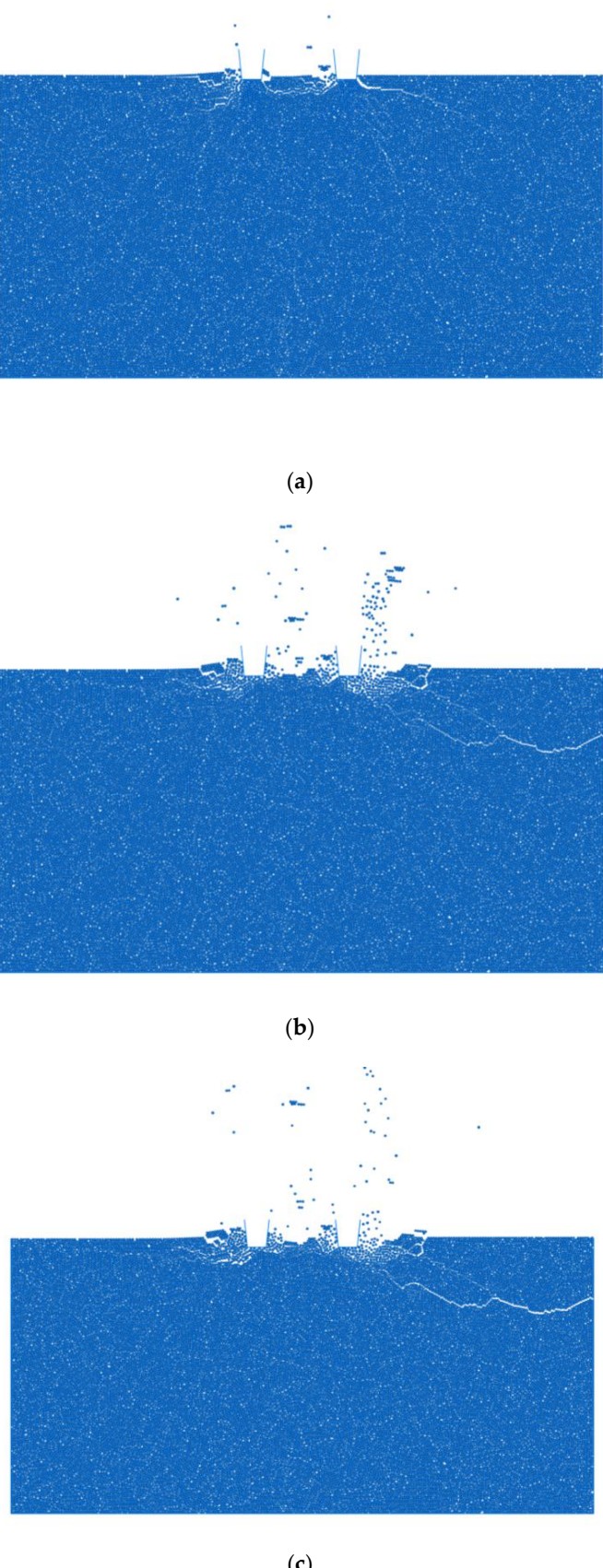

**Figure 21.** Development of the crushed zones and cracks after cutting with two disc cutters at a spacing of 62.5mm: (**a**) penetration of 3.2mm; (**b**) penetration of 4.4mm; (**c**) penetration of 6.4mm.

In this research, we compared two different dimensional DEM models of TBM disc-cutter rock fragmentation and analyzed the advantages and disadvantages of the two-dimensional and three-dimensional numerical models. However, modeling the TBM disc cutter when cutting rock is still in its infancy, and there are still many issues that need to be further studied. For example, because of the existing computational limits, PFC models cannot be used for the simulation of full-scale tests. This study's numerical model only simulated part of the process of disc-cutter rock fragmentation. In order to fully reproduce the process of the TBM cutting rock, the numerical simulation computational power needs to be improved. On the other hand, natural rocks have complex structures and contain joints and cracks. These components of natural rocks are hard to simulate in numerical simulation models. This is also an important reason for the difference between the numerical simulation results and experimental results. Therefore, it is necessary to carry out further and more accurate numerical studies on rock structures in detail.

## 5. Conclusions

Discrete element models in PFC$^{3D}$ and PFC$^{2D}$ were used to simulate rock fragmentation when using disc cutters in different configurations. FJCM was used to simulate the rock specimen and offer more realistic fracturing behavior in the context of rock excavation, where the contact parameters could be better controlled to show the behavior of rock, as measured by typical rock mechanics testing. The cutting force calculated under various conditions could be used to calculate the SE for various disc-cutter spacings and penetrations. The numerical simulation results were verified in two stages, one by comparing the calculated results with the results of rock mechanics testing, thus determining the mesoscopic parameters for the PFC models. This level of verification was used to compare the results of the numerical analysis with full-scale rock-cutting tests using LCM data for a selected granite (CRG) at the CSM rock excavation laboratory. The recorded force in the LCM tests was slightly lower than the forces calculated in the numerical simulations. The rolling coefficients of RC and SE showed the same trends as the LCM tests, which validates the numerical simulation results.

The differences between the two- and three-dimensional numerical models were also examined. While the two-dimensional numerical model allowed for tracking the crack development, the calculated values were different from the full-scale test results. This means that they could be used for the qualitative evaluation of the fracturing process. Three-dimensional PFC models were more realistic and could be used to estimate the forces and SE. The results of this research could help in predicting the cutting forces and, hence, in estimating the performance of TBMs in specific rock conditions. Furthermore, the model can be used in the future to simulate rock anisotropy and jointing, to allow for the estimation of cutting forces, SE, and machine performance under more realistic natural conditions. Besides this, the modeling can also assist in improvements in disc-cutter design.

Due to the existing computational limits, PFC models cannot be used for the simulation of full-scale tests, such as LCM testing. However, these limitations could be overcome by using additional computational power, which would permit the simulation of more complex cutting conditions that could help in making further advances in rock-cutting technology.

**Author Contributions:** C.X.: Conceptualization, methodology, writing—original draft. Y.Z.: Data curation, software. D.S.: Writing—review and editing. X.G.: Supervision, writing—review. X.L.: Funding acquisition, resources, validation, writing—review and editing. E.W.: Supervision, writing—review. R.L.: editing. All authors have read and agreed to the published version of the manuscript.

**Funding:** Supported by the National Natural Science Foundation of China (52109125 and 52090081), the Open Research Fund Program of the State Key Laboratory of Hydroscience and Engineering (sklhse-2022-C-04, sklhse-2022-D-01), the Natural Science Foundation of Hubei Province, China (ZRMS2022000712), the Open Research Fund of the SINOPEC Key Laboratory of Geophysics (WX2021-01-12), the China Postdoctoral Science Foundation (2020M680583), and the National Postdoctoral Program for Innovative Talent in China (BX20200191).

**Institutional Review Board Statement:** Not applicable.

**Informed Consent Statement:** Not applicable.

**Data Availability Statement:** Not applicable.

**Conflicts of Interest:** The authors declare that they have no known competing financial interests or personal relationships that could have appeared to influence the work reported in this paper.

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
