# Peer review of "Comparison of DEM Models with Different Elemental Dimensions for TBM Disc Cutter Rock Fragmentation"

_sustainability, doi:10.3390/su141912909_

Round 1
Reviewer 1 Report
The article contains interesting numerical tests relating to selected parameters of cutting with mining discs. Obtaining appropriate progress in specific geological and mining conditions determines the timely completion of the excavation project, which is why the technologies of driving, and in particular the geometry of the cutting tools, are constantly being improved. Below are some comments and suggestions:
1. The introduction should include information on laboratory tests (two / three) in which new design solutions of cutting tools are tested.
2. The second chapter presents numerical programs for modeling the states of stress/strain in the rock mass. And were any numerical programs used for the disc cutter that could be used to determine the stress distribution on its circumference?
3. In the third chapter, it should be written whether the numerical tests of the cuts were made parallel to the stratification or perpendicular?
4. In the subsection 3.2 the description of full-scale rock cutting tests should be improved; first of all, it would be useful to take a photo of the laboratory stand with measuring equipment.
5. In the fourth chapter on the discussion of results, (two/three) references should be added relating to tests performed in other research centers where the cutter disc is tested or modeled.
Reviewer 2 Report
Please find my comments in the attachment.
